# Associations between First Trimester Maternal Nutritional Score, Early Markers of Placental Function, and Pregnancy Outcome

**DOI:** 10.3390/nu12061799

**Published:** 2020-06-17

**Authors:** Francesca Parisi, Valeria M. Savasi, Ilenia di Bartolo, Luca Mandia, Irene Cetin

**Affiliations:** 1Department of Woman, Child and Neonate, Buzzi Children Hospital, ASST Fatebenefratelli Sacco, 20154 Milan, Italy; ileniadibartolo@hotmail.it (I.d.B.); irene.cetin@unimi.it (I.C.); 2Department of Woman, Child and Neonate, Luigi Sacco Hospital, ASST Fatebenefratelli Sacco, 20154 Milan, Italy; valeria.savasi@unimi.it (V.M.S.); lucamandia@gmail.com (L.M.); 3Department of Biomedical and Clinical Sciences, Università degli Studi di Milano, 20122 Milan, Italy

**Keywords:** healthy diet, maternal nutrition, placental volume, first trimester placental markers, preterm delivery

## Abstract

This study investigated the associations between maternal adherence to a healthy diet, first trimester placental markers, and pregnancy outcome. Singleton spontaneous pregnancies were enrolled at 11^+0^–13^+6^ gestational weeks in a prospective cohort study. A nutritional score (0–10) measuring the adherence to a healthy diet was calculated. A transabdominal ultrasound scan for placental marker assessment was performed (uterine artery (UtA) doppler, placental volume). Biochemical placental markers were recorded (Pregnancy Associated Plasma Protein A (PAPP-A), free β- Human Chorionic Gonadotropin (HCG)). Birth outcomes were obtained from medical records. Associations between the maternal nutritional score, first trimester placental markers, and pregnancy outcome were investigated by using multi-adjusted general linear models. In total, 112 pregnancies were enrolled with a median nutritional score of 7 (range 3–10). Median gestational age at birth was 277 days (range 203–296). The nutritional score was positively associated with PAPP-A concentrations, whereas a negative association was detected with the UtA mean pulsatility index and placental volume. A positive association was detected between nutritional score and gestational age at birth. This study demonstrates that a first trimester nutritional score as a measure of adherence to a healthy diet is significantly associated with early biochemical and ultrasound markers of placental development, with further association with gestational age at birth.

## 1. Introduction

In the age of the obesity pandemic, an evolutionary revolution is underway in reproductive medicine: Maternal nutrition in pregnancy, as the main determinant of fetal nutrition, may lead to permanent modifications in fetal growth trajectories, gene expression, and metabolic pathways, which eventually will turn into a modified disease risk profile in postnatal life [1,2]. In the last decades, the focus of research has additionally moved to the preconceptional period and the first trimester of pregnancy (the so-called periconceptional period). Besides the well-known association with gamete competence, reproductive outcome, and embryonic morphological development, periconceptional maternal nutritional adequacy is essential for a normal adaptation to pregnancy and a proper placentation, leading to overt diseases in the second half of pregnancy and increased risk of chronic disease in adulthood in case of an unbalance [2,3]. In fact, several micronutrients (e.g., antioxidants, one-carbon biomarkers, vitamin D) are important regulators of endothelial function, angiogenesis, and placental metabolic functions, thus explaining the association between micronutrient deficiencies in the periconceptional period, placental oxidative stress, and subsequent pregnancy complications, including preterm delivery and hypertensive disorders [4,5]. As single food items and micronutrients, without considering the composite biological interactions between nutrients, have shown inconclusive associations with complex health outcomes, recent research took into consideration maternal dietary patterns as potential predictors of adverse pregnancy and health outcomes [6]. In particular, low adherence to the Mediterranean diet has been associated with several adverse outcomes, both in pregnancy (e.g., birth defects, gestational diabetes mellitus, hypertensive disorders) and childhood (e.g., atopy, cardiovascular risk profile, body composition) [7,8,9].

The aim of this study was to evaluate the associations between a maternal nutritional score measuring the adherence to a healthy diet, first trimester biochemical and ultrasound markers of placental function, and pregnancy outcome.

## 2. Materials and Methods

### 2.1. Study Population

The present study was conducted at the Department of Woman, Child, and Neonate of the Luigi Sacco University Hospital, Milan, Italy, as the pilot study of an ongoing multicenter prospective study started in 2018. The protocol was approved by the Medical Ethical and Institutional Review Board at the Luigi Sacco University Hospital in Milan, Italy, and all participants signed a written informed consent form before participation (approval number 48, 2013).

Between February and June 2017, healthy women of at least 18 years of age, carrying a singleton ongoing pregnancy, and undergoing the first trimester combined screening test for aneuploidies between 11^+0^ and 13^+6^ weeks of gestation were eligible for enrollment. All patients were recruited at the time of the ultrasound scan. The first trimester combined screening is offered to all pregnant women in Italy, independently of their a priori risk. Ongoing pregnancy, spontaneous conception, and no known maternal pregestational pathologies or therapies were inclusion criteria. Multiple pregnancy, conception through assisted reproduction techniques, ectopic implantation, or any congenital anomaly further confirmed at birth were exclusion criteria. Gestational age was defined from the last menstrual period confirmed by an ultrasound crown-rump length (CRL) measurement within seven days between 11^+0^ and 13^+6^ weeks. In the case of discrepancy ≥seven days, gestational age was based on the CRL measurement according to the Robinson curves [10].

### 2.2. Data Collection

At enrollment, all women filled a general questionnaire, including family, obstetric, and medical history; anthropometric measurements; and details on age, ethnicity, education, and periconceptional lifestyle and supplementation. Data on birth outcomes, including gestational age, mode of delivery, blood loss, birth weight, newborn sex, and pregnancy outcome, were recorded from a medical registry or phone interview.

As required by the first trimester combined screening test, one venous blood sample per pregnancy was collected at 10 weeks of gestation for routine determination of serum pregnancy-associated plasma protein-A (PAPP-A) and free β-human chorionic gonadotropin (β-HCG) by using a solid-phase two-site sequential chemiluminescent immunometric assay (BRAHMS Kryptor, Hennigsdorf, Germany).

All women received a transabdominal three-dimensional (3-D) ultrasound scan between 11^+0^ and 13^+6^ weeks of gestation with a 3–5 MHz probe of a Samsung Healthcare equipment WS80 (Samsung Electronics, Milan, Italy). All ultrasound measurements were performed by a single Fetal Medicine Foundation certified sonographer (L.M.). Beside ultrasound parameters required for the combined screening test (CRL, nuchal translucency), transabdominal measurements of Doppler velocimetry of uterine arteries (UA) and 3-D placental volume were performed for all participants. UTA Doppler velocimetry was obtained after identifying each UTA along the side of the internal cervical os. Pulsed wave Doppler was used with a 2-mm sampling gate and an insonation angle <30°. The mean pulsatility index (PI) was measured on at least three similar consecutive waveforms [11].

The 3-D volume of the placenta was acquired and stored for off-line analysis by virtual organ computer-aided analysis (VOCAL). Placental localization was defined on a two-dimensional real-time scan. Three-dimensional scans were performed at the maximum scanning angle with the probe perpendicular to the placental plate. A sequence of 6 sections of the placenta were obtained at a 30° rotation distance between each other. The manual contouring of the placenta was performed in each plane and the placental volume was automatically calculated by using the VOCAL program. Datasets with insufficient quality to clearly define the placental contours were excluded from the analysis. All placental volume measurements were performed three times by one researcher (I.B.) and the mean value was considered for the analysis.

All women completed an adapted version of the nutritional questionnaire developed and based on the International Federation of Gynecology and Obstetrics (FIGO) recommendations on adolescent, preconception, and maternal nutrition [12,13] (Table 1).

The table lists the questions based on the FIGO recommendations on maternal nutrition and measuring the adherence to a healthy diet and lifestyle in pregnancy (12). A one-point is calculated in case of affirmative answer for: Consumption of meat 2–3 times per week, fruit and vegetables at least 5 times per day, fish 1–2 times per week, dairy products daily, whole cereals at least once per day, sweet and snacks less than 5 times per weeks, first trimester hemoglobin concentrations higher than 110 g/L, folic acid supplementation, use of iodized salt, and sun exposure at least 10–15 min per day. The absolute frequencies and percentages of the study population are shown.

The questionnaire included six binary questions (yes/no) on the consumption frequency of: Meat (two to three times per week), fruit and vegetables (at least five portions per day), fish (once to twice weekly), dairy products (daily), whole grain (at least once per day), sweet and snacks (less than five times per week), and three binary questions on folic acid supplementation, sun exposure (at least 10 to 15 min per day), and hemoglobin concentrations (higher than 110 g/L). Additional adaptations of the questionnaire were based on the Italian guidelines on maternal nutrition during pregnancy, including one question on the consumption of iodized salt and the modified recommended intake of fruit and vegetables to five portions per day. The questionnaire provided a final calculation of a 0 to 10 score, measuring the adherence to a healthy diet and lifestyle in pregnancy.

### 2.3. Statistical Analysis

Maternal baseline characteristics, biochemical and ultrasound markers of placental function, and birth outcomes were described as a median and range for quantitative variables, and absolute and relative frequencies for categorical variables. A first trimester nutritional score representing the adherence to a healthy diet and lifestyle was calculated as the sum of 10 questions, as described in Table 1. Log10 transformation of non-normally distributed variables was performed to approximate Gaussian distributions. Bivariate correlations were performed to investigate the associations between maternal baseline characteristics and the first trimester nutritional score. In order to investigate associations between pregestational maternal nutritional status and the study outcomes, maternal pregestational BMI was firstly evaluated in association with first trimester placental markers and birth outcomes in a multi-adjusted general linear model. General linear models adjusted for confounding factors (gestational age at enrollment, maternal age, parity, body mass index (BMI), smoking habit, fetal sex) were secondly performed to investigate associations between the nutritional score (independent variable), first trimester biochemical (PAPP-A, free β-HCG), and ultrasound (mean UtA PI, placental volume) markers of placental function, and pregnancy outcomes (mode of delivery, gestational age at birth, birth weight, blood loss) (dependent variables). Finally, Chi square and relative risk calculation were performed to evaluate the relative risk of spontaneous preterm delivery, gestational diabetes mellitus, hypertensive disorders, and small and large for gestational age babies (birth weight <10th percentile or >90th for gestational age, respectively) in women with a first trimester nutritional score <5 compared to score ≥5.

*p*-values < 0.05 were considered statistically significant. All analyses were performed using SPSS Statistics for Windows, Version 21.0 (IBM Corp. Armonk, New York, NY, USA) and R version 3.2.1 (The R Foundation for Statistical Computing).

## 3. Results

In total, 112 healthy women carrying a singleton spontaneous pregnancy were included in the analysis after excluding pregnancies achieved after assisted reproductive techniques (*n* = 5), fetal or neonatal congenital anomalies (*n* = 3), and pregestational maternal pathologies (*n* = 9). No differences were detected in the nutritional score distribution between pregnancies excluded for congenital anomalies and the included study population. The median gestational age at recruitment was 12^+4^ weeks (range 11^+0^–13^+6^) with a median CRL of 63.2 mm (range 45.0–84.0).

Table 2 shows the maternal baseline characteristics and first trimester markers of placental function in the study population. The study population included 11 patients who were underweight and seven who were obese. A negative correlation was detected between pregestational BMI and gestational weight gain at enrollment. Placental volume could be assessed as high quality in 65 out of 98 datasets (measurement success rate 66%).

The study population included 112 women carrying a singleton pregnancy with a non-malformed outcome. Gestational weight gain was defined at the enrollment. 

The median first trimester nutritional score at enrollment was 7 (range 3–10) (Table 1). A significant proportion of the study population (28%) showed a nutritional score lower than five. In particular, the consumption of fruit and vegetables, whole grains, and sweets and snacks was in line with the international recommendations in 35.7%, 25.9%, and 54.4% of the study population, respectively. The nutritional score was positively correlated with maternal age (*r* = 0.20; *p* = 0.04), whereas no correlations were detected with pregestational BMI, gestational weight gain at enrollment, parity, smoking habit, and geographical origin. No differences in pregestational BMI, hemoglobin concentrations, and gestational weight gain distribution were detected between the two subgroups defined by nutritional scores <5 and ≥5.

Maternal pregestational BMI was positively associated with PAPP-A concentrations (β = 0.11, *p* = 0.01) and placental volume (β = 2.3, *p* = 0.002), and negatively associated with the CRL/placental volume ratio (β = −0.04, *p* = 0.01), whereas no associations were detected with birth outcomes, including birthweight and gestational age at birth.

Table 3 lists the birth outcomes of the study population. Pregnancy outcomes could be recorded for 94 patients out of 112. Nine babies were classified as large for gestational age (9.6%) and three as intrauterine growth restricted (3.2%). One very preterm spontaneous delivery was recorded (29 weeks of gestation), whereas five babies were delivered between 32^+0^ and 37^+0^ weeks of gestation. Three pregnancies were diagnosed with gestational hypertension and underwent vaginal delivery at term. In total, 22 women out of 94 with a known birth outcome underwent a cesarean section (23%) and 29 an induction of labor (30%).

Pregnancy outcomes were recorded for 94 women out of 112.The table lists the median and range for quantitative variables and the absolute and relative frequencies for categorical variables. 

Table 4 shows the results from the crude (adjusted for gestational age at recruitment) and fully adjusted general linear models.

Effect estimates represent the amount of change in first trimester placental markers per unit of increase in the nutritional score. The crude model is adjusted for the gestational age at the time of the ultrasound scan. The analysis on birth weight is additionally adjusted for gestational age at birth in both crude and fully adjusted models. The fully adjusted model is further adjusted for maternal BMI, age, smoking habit, geographical origin, parity, and fetal sex. 

In the fully adjusted model, the first trimester maternal nutritional score was positively associated with PAPP-A concentrations, whereas a negative association was detected with UtA mean PI and placental volume. No significant associations were detected for free β-HCG. The CRL/placental volume ratio was positively associated with the nutritional score, with a *p*-value close to significance after full adjustment. No associations were detected between single nutritional items and placental biochemical and ultrasound markers (data not shown).

The analysis on birth outcomes showed a significant positive association between the first trimester nutritional score and gestational age at birth in the fully adjusted model, whereas no associations were detected with birth weight, blood loss, and the APGAR score. Women with a first trimester nutritional score lower than five showed a significant increase in preterm deliveries, with a relative risk of 1.44 (95% CI: 1.25–1.65). No associations were detected with hypertensive disorders of pregnancy, gestational diabetes mellitus, intrauterine growth restriction, large for gestational age babies, mode of delivery, and post-partum hemorrhage. A significant positive association was lastly detected between first trimester placental volume and birthweight in a fully adjusted model (β = 5.07, *p* = 0.02).

## 4. Discussion

This study demonstrates that a first trimester nutritional score measuring the adherence to a healthy diet and lifestyle in pregnancy is significantly associated with early biochemical and ultrasound markers of placental development independently of pregestational BMI. In particular, higher first trimester maternal nutritional scores were associated with increased PAPP-A concentrations, lower UtA mean PI, and decreased placental volumes at the first trimester screening ultrasound, whereas no associations were detected with free β-HCG. The obtained measurements of placental volume were comparable with previous data and small variations could be firstly dependent on the mean gestational age at the ultrasound scan, as the placenta more than doubles its volume starting from 11 to 14 weeks [14,15]. The analysis on birth outcomes additionally showed a positive association between the maternal nutritional score and gestational age at birth (*p* = 0.05). Despite the small number of adverse outcomes not allowing speculations in this sense, a significant increase in the risk of preterm birth seems to be associated with low nutritional scores, whereas no associations were found with birth weight, intrauterine growth restriction, gestational diabetes mellitus, large for gestational age babies, and hypertensive disorders of pregnancy after full adjustment. In line with previous results, the single food items of the score calculation, without considering the composite interactions between food groups and nutrients, were not associated with both early placental markers and complex pregnancy outcomes [6]. Lastly, the present study provides a picture of the first trimester maternal nutritional habits in a cohort of women in Milan, showing an alarming low adherence to the international nutritional recommendations for a healthy pregnancy.

The first trimester maternal nutritional score was positively correlated with maternal age, whereas no correlations were detected with pregestational BMI. In line with a negative correlation detected between maternal BMI and gestational weight gain, this result may underline a positive effect of early nutritional counselling addressing women with an abnormal BMI as required by local obstetric care protocols in the improvement of nutritional habits during pregnancy.

The present study firstly shows a positive association between the first trimester maternal nutritional score and early biochemical and ultrasound markers of placental function. In line with our results, homologous and heterologous fertility treatments have recently been associated with significantly lower first trimester PAPP-A and placental growth factor levels, higher free β-HCG concentrations, and modified mean UtA PI according to fertility treatments [16,17]. Several other maternal characteristics, including height, weight, age, ethnicity, parity, and pregestational diabetes, have been strongly associated with first trimester serum biomarkers of placental function, further affecting the risk of placenta-related pathologies in the second half of pregnancy [18,19]. Moreover, maternal periconceptional exposures, including smoking habit and alcohol use, were studied in both animal and human models of placental development [3,20,21,22]. In particular, maternal smoking was found to decrease both PAPP-A and free β-HCG concentrations, with no significant effects on the first trimester human placental volume [23,24,25]. All these data, together with our results, underline the importance of maternal baseline characteristics, habits, and mode of conception on first trimester markers of placental development, explaining the subsequent association with maternal maladaptation to pregnancy, placental dysfunction, and overt disease in the second half of pregnancy. In this scenario, a reduced first trimester placental volume has been shown to be predictive of preeclampsia and low birthweight [26,27]. Despite the present study showing a negative association between the maternal nutritional score and first trimester placental volume, a positive association close to significance was detected between the nutritional score and the CRL/placental volume ratio, as to indicate a smaller but more efficient first trimester placenta in the case of higher nutritional scores. The positive association detected between first trimester placental volume and birthweight further corroborates this hypothesis. Conversely, maternal pregestational BMI showed an independent positive association with placental volume and a negative association with the CRL/placental volume ratio, thus suggesting larger but less efficient placentas in the case of a higher BMI.

Maternal malnutrition represents another well-known risk factor for impaired maternal adaptation to pregnancy, placentation, and intrauterine fetal growth [3,28]. Several nutritional factors, including micronutrient and folic acid supplementation and the adherence to the Mediterranean diet, were studied in association with second and third trimester markers of placental function (umbilical artery and UtA Doppler indices, serum biomarkers Placental Growth Factor (PlGF) and Soluble fms-like tyrosine kinase-1 (sFLT1)), showing decreased vascular resistance, increased angiogenic marker concentrations, and increased placental weight in case of supplementation and high adherence to a healthy diet [5,7,29,30]. Again, maternal obesity was studied in association with second and third trimester ultrasound and biochemical markers of placental function, showing larger placental volume, decreased angiogenic factors, increased anti-angiogenic factor concentrations, and increased placental weight at birth [31,32,33]. Moving towards the first trimester of pregnancy, early nutritional exposures, including low adherence to a dietary pattern high in fish and olive oil and low in meat intake and deranged one-carbon biomarkers, have recently been associated with embryonic morphological developmental delays, first trimester cerebellar size, and decreased embryonic growth trajectories, reversing the idea that all embryos must grow and develop with the same time and velocity [34,35,36]. To our knowledge, no data have been previously published on the association between maternal adherence to a healthy diet and first trimester placental markers. Maternal nutrition may impact early placentation by modifying phospholipid, protein, and DNA synthesis; by modulating both systemic and local oxidative stress and inflammatory responses; and finally by affecting angiogenesis and altering placental epigenetics [22,37,38].

The present study shows several strengths and limitations. Despite it being conceived only as a pilot study of a larger Italian multicenter study, the small sample size still represents the main limitation of the study. In particular, due to the very small number of adverse outcomes, the analysis on birth outcomes should be interpreted only as an indication of a possible association that needs to be confirmed on a larger scale of low- and high-risk pregnancies. Furthermore, gestational weight gain at term, as well as family or personal history of obstetric complications were not recorded, and we cannot exclude an independent effect of these parameters on pregnancy outcome. Moreover, second and third trimester ultrasound or biochemical data of placental and fetal development were not available. The nutritional score was calculated in the late first trimester, when modifications of nutritional behaviors could not be excluded due to nausea or vomit. Subsequent changes in nutritional habits, exposures, and supplementation could impact on the observed pregnancy outcomes. Lastly, energy intake, as a crucial component of nutritional habits and risk, was unknown.

On the other hand, this study presents several strengths. The included population represents a well-defined sample of healthy women with a low-risk pregnancy. The inclusion of underweight and obese women further increases the external validity of our findings, being in line with the reported national prevalence of abnormal BMI among fertile women. All ultrasound measurements were performed by the same certified sonographer, thus reducing the interobserver variability. The nutritional questionnaire developed and based on the FIGO recommendations on adolescent, preconception, and maternal nutrition represents an important tool providing information on nutritional habits during pregnancy in a reproducible and fast way in clinical settings. Although residual confounding could not be completely excluded, full adjustment for all known factors associated with placental growth and development were considered in the final model.

Placenta-related pregnancy complications, including preterm birth, hypertensive disorders, and fetal growth restriction, represent the most relevant contributor to maternal and neonatal morbidity and mortality and are known to be strongly associated with maternal nutritional behaviors [2,39,40]. First trimester maternal serum biomarkers and uterine artery doppler indices have shown promising capacities in the prediction of early and severe preeclampsia, as well as interesting associations with further obstetric complications, including gestational diabetes and intrauterine growth restriction, thus explaining the relevance of investigating environmental factors and exposures able to impact on early markers of embryonic and placental dysfunction [21,41,42,43,44,45]. This knowledge could provide more and more precise prediction models of clinical diseases and pregnancy outcome, improve placental development, and develop preventive strategies to finally improve the health outcome of future generations [21,42,43,44,45].

This study provides the first evidence of an association between maternal nutritional habits, first trimester markers of placental function, and gestational age at birth, laying the foundations for future intervention strategies starting as early as the first trimester of pregnancy. Moreover, the study shows the utility of a very easy tool of nutritional screening in order to evaluate the most relevant nutritional deficiencies in pregnancy, with impacts on the first stages of placentation and pregnancy outcome. The larger ongoing multicenter Italian study on both low- and high-risk pregnancies will provide a more accurate picture of the current nutritional habits in pregnancy, as well as more reliable evidence of the detected association.

## Figures and Tables

**Table 1 nutrients-12-01799-t001:** Periconceptional nutritional score as a measure of maternal adherence to a healthy diet and healthy lifestyle.

Score Component	Frequency
Meat, yes *n* (%)	90 (80.4)
Fruit and Vegetables, yes *n* (%)	40 (35.7)
Fish, yes *n* (%)	79 (70.5)
Dairy, yes *n* (%)	79 (70.5)
Whole cereals, yes *n* (%)	29 (25.9)
Sweets and Snacks, yes *n* (%)	61 (54.5)
Hemoglobin, yes *n* (%)	91 (81.3)
Folic acid, yes *n* (%)	109 (97.3)
Iodized salt, yes *n* (%)	74 (66.1)
Sun exposure, yes *n* (%)	69 (61.6)

**Table 2 nutrients-12-01799-t002:** Maternal baseline characteristics and first trimester placental markers of the study population (*n* = 112).

**Maternal Baseline Characteristics**	
Age, y, median (range)	31 (18–43)
Geographical origin, Western, *n* (%)	96 (85.7)
Employment status, yes *n* (%)	84 (75.0)
BMI, kg/m^2^, median (range)	21.9 (15.6; 39.5)
Gestational weight gain, kg, median (range)	1 (−5; 8)
Nulliparous, *n* (%)	65 (58.0)
Periconceptional smoking, *n* (%)	13 (11.6)
Hemoglobin concentration, g/l, median (range)	13.1 (10.0; 15.3)
**Placental Ultrasound and Biochemical Markers**	
PAPP-A, mIU/ml, median (range)	2.73 (0.31; 14.86)
PAPP-A, MoM, median (range)	1.20 (0.29; 4.60)
free β –HCG, ng/ml, (median, range)	41.5 (4.6; 373.0)
free β –HCG, MoM, (median, range)	1.12 (0.29; 3.31)
Placental volume, ml, median (range)	51.1 (36.9; 96.5)
UtA mean PI, median (range)	1.45 (0.58; 2.91)

BMI: body mass index, UtA: uterine artery, PI: pulsatility index, PAPP-A: pregnancy-associated plasma protein-A, free β-HCG: free β-human chorionic gonadotropin, MoM: multiple of median, GA: gestational age, IUGR: intrauterine growth restriction, GDM: gestational diabetes mellitus.

**Table 3 nutrients-12-01799-t003:** Pregnancy and birth outcomes of the study population.

Birth Outcomes	
GA at birth, days, median (range)	277 (203; 296)
Fetal sex, male *n* (%)	55 (49.1)
Birth weight, g, median (range)	3285 (1180; 4360)
Blood loss, mL, median (range)	325 (50; 4000)
Postpartum hemorrhage, *n* (%)	12 (12.7)
Spontaneous preterm delivery <37 weeks, *n* (%)	6 (6.4)
Hypertensive disorders, *n* (%)	3 (3.2)
IUGR, *n* (%)	3 (3.2)
GDM, *n* (%)	11 (11.7)

GA: gestational age, IUGR: intrauterine growth restriction, GDM: gestational diabetes mellitus.

**Table 4 nutrients-12-01799-t004:** Results from multi-adjusted general linear models on the associations between nutritional score (independent variable), first trimester biochemical and ultrasound markers of placental function, and pregnancy outcomes (dependent variables).

	Crude Model β (95%CI)	*p*-Value	Fully Adjusted Model β (95%CI)	*p*-Value
PAPP-A, mIU/mL	**0.27 (0.00; 0.31)**	**0.05**	**0.26 (0.02; 0.50)**	**0.05**
Free β-HCG, ng/mL	−0.63 (−6.29; 5.02)	0.82	−0.70 (−6.51; 5.09)	0.80
Placental Volume, mL	**−3.06 (−7.86; −1.74)**	**0.04**	**−3.31 (−7.93; −1.11)**	**0.04**
UtA Mean PI	−0.02 (−0.07; 0.01)	0.06	**−0.01 (−0.07; −0.00)**	**0.05**
CRL/Placental Volume	0.08 (−0.05; 0.20)	0.09	0.10 (−0.02; 0.22)	0.06
GA at Birth, days	1.4 (0.0; 3.0)	0.06	**1.4 (0.3; 3.2)**	**0.05**
Birth Weight, g	−25.5 (−77.2; 26.1)	0.32	−18.7 (−73.2; 35.8)	0.50

PAPP-A: pregnancy-associated plasma protein-A, free β-HCG: free β-human chorionic gonadotropin, UtA: uterine artery, PI: pulsatility index, CRL: crown-rump length, GA: gestational age.

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
