# Peer review of "Associations between First Trimester Maternal Nutritional Score, Early Markers of Placental Function, and Pregnancy Outcome"

_nutrients, 2020, doi:10.3390/nu12061799_

Round 1
Reviewer 1 Report
The authors try to address the question, whether nutrition on "Mediterranian Diet" is adequate to improve the course and outcome of pregnancy. This is a serious and ambitious task. The paper is - regarding the topic and cited references - well written. However, there are several major queries and objections:
Major:
Line 70: as preterm delivery is included, there is no reason, why miscarriage and intrauterine death should be excluded.
L. 108f/Table 1: The table and text do not provide sufficient specific information. Please present the whole questionnaire! I really do not see, how the topics in Table 1 will define a mediterranean diet? Several meals with fish per week are characteristic for Baltic Sea neighbours as well. What is specific for "Mediterranean"?
Table 2: A BMI <18.5 is defined as underweight, while >30 is defined as obesity. Underweight is per se characterized by nutrition deficits, overweight by increased risk of preterm delivery. Were the under- and overweight persons equally distributed along the scaling? Who are those loosing or gaining too much weight during pregnancy? Are they equally distributed in the <5/>5 scale? A hemoglobin of 10 is per se an indicator of undernutrition. Was this value equally distributed along the scale from 3 - 10?
L. 192f: Are all and particularly these data corrected for BMI? Was the BMI identical in <5/>5 nutrition score? Were the premies from low BMI mothers?
L. 198: How dis lifestyle and BMI correlate with the nutrition score? Was there an effect of BMI on any outcome parameter? Was it corrected for?
L. 213: Actually, the reviewer does not see "a detailed picture of the first trimeter maternal nutritional habits".
L. 214: The reviewer questions the unrestricted health status of the heavy under- and overweight persons.
L. 269; homogenous: This group of women is not homogenous with severe under and overweight included, where no data are provided in relation to the diet score. Quetion is, whether the collective is representative for the overall population.
Minor:
Line 101-103: How many data sets were excluded?
L. 135: the correct biological term is sex rather than gender. Please change troughout.
L. 205f: leave out the word "significant" and include p-value.
L. 280ff: A sentence of more than 7 lines may be a bit too long.
Author Response
Reviewer 1
Comments and Suggestions for Authors
The authors try to address the question, whether nutrition on "Mediterranian Diet" is adequate to improve the course and outcome of pregnancy. This is a serious and ambitious task. The paper is - regarding the topic and cited references - well written. However, there are several major queries and objections:
Major:
Line 70: as preterm delivery is included, there is no reason, why miscarriage and intrauterine death should be excluded
We thank the reviewer for this important comment and we agree with her/him about the relevance of this topic and eventual inclusion of miscarriage/IUFD in order to evaluate associations between maternal nutritional habits and pregnancy outcome. Actually, the present study includes women enrolled at the time of the ultrasound scan of the combined test (11+0-13+6 weeks) with ongoing pregnancy as inclusion criterion. If miscarriage was diagnosed at enrollment, the patient was excluded also due to the difficulties related to the study proposal to the woman who received this unfortunate diagnosis. This explains why we did not collect the nutritional questionnaire for two women with miscarriage diagnosed at 12+1 and 11+5 gestational weeks, respectively. Furthermore, no IUFD were detected in the whole study population. As we understand the reviewer's comment, we modified line 70 as follows: " Ongoing pregnancy, spontaneous conception and no known maternal pregestational pathologies or therapies were inclusion criteria. Multiple pregnancy, conception through assisted reproduction techniques, ectopic implantation, or any congenital anomaly further confirmed at birth were exclusion criteria."
- 108f/Table 1: The table and text do not provide sufficient specific information. Please present the whole questionnaire! I really do not see, how the topics in Table 1 will define a mediterranean diet? Several meals with fish per week are characteristic for Baltic Sea neighbours as well. What is specific for "Mediterranean"?
We definitely agree with the reviewer. Despite the Mediterranean diet is considered the healthy diet mostly represented in Italy and the questionnaire has been adapted based on the National guidelines related to the adherence to this specific dietary pattern, the nutritional checklist is too generic to define a Mediterranean dietary pattern. Therefore, we changed "Mediterranean" with "healthy" diet throughout the manuscript. Furthermore, the M&M section has been adapted as required and the questionnaire has been further explained in the text.
Table 2: A BMI <18.5 is defined as underweight, while >30 is defined as obesity. Underweight is per se characterized by nutrition deficits, overweight by increased risk of preterm delivery. Were the under- and overweight persons equally distributed along the scaling? Who are those loosing or gaining too much weight during pregnancy? Are they equally distributed in the <5/>5 scale? A hemoglobin of 10 is per se an indicator of undernutrition. Was this value equally distributed along the scale from 3 - 10?
We appreciate this comment. The study population includes 11 patients being underweight and 7 obese. The fully adjusted model always includes adjustment for pregestational BMI, thus reducing the risk of bias. Pregestational BMI, as well as weight gain at recruitment were not correlated with the nutritional score, in contrast to maternal age. No differences in maternal pregestational BMI, hemoglobin concentrations, and gestational weight gain distribution were detected between the two groups defined by nutritional scores higher and lower than five. The Results section has been modified including this information.
- 192f: Are all and particularly these data corrected for BMI? Was the BMI identical in <5/>5 nutrition score? Were the premies from low BMI mothers?
We confirm that the fully adjusted model always includes adjustment for pregestational BMI, besides gestational age, maternal age, smoking habit, geographical origin, parity and fetal sex. The indicated line has been modified accordingly. As specified above, no differences in pregestational BMI were detected according to the two subgroups defined by nutritional score, which can be probably related to the specific nutritional counselling received by women with pathological BMI as required by local obstetric care protocols. This may also explain the negative correlation between pregestational BMI and gestational weight gain. The analysis of an independent effect of pregestational BMI on birth outcomes, including birthweight and gestational age at birth, showed no significant results. We added this information in the Results section.
- 198: How dis lifestyle and BMI correlate with the nutrition score? Was there an effect of BMI on any outcome parameter? Was it corrected for?
In a multi-adjusted model, maternal BMI was positively associated with PAPP-A (beta= -0.11, p=0.01) and placental volume (beta= -2.3, p=0.002), whereas a negative association was detected with CRL/placental volume ratio (beta= -0-04, p=0-01). As previously specified, BMI has been always included for adjustment in the fully adjusted model (table 4), so that the association between nutritional score and the study outcomes includes the independent effect of pregestational BMI. These results have been added to the specific section according to the reviewer's suggestions.
- 213: Actually, the reviewer does not see "a detailed picture of the first trimeter maternal nutritional habits".
We agree with the reviewer and modified the sentence.
- 214: The reviewer questions the unrestricted health status of the heavy under- and overweight persons.
We agree with the reviewer and modified the sentence accordingly.
- 269; homogenous: This group of women is not homogenous with severe under and overweight included, where no data are provided in relation to the diet score. Quetion is, whether the collective is representative for the overall population.
We agree with the reviewer and the word 'homogeneous' has been deleted. Nevertheless, we still think that the inclusion of underweight and obese women further increases the external validity of our findings, being in line with the reported national prevalence of abnormal BMI among fertile women and so more representative of the Italian general population. The Discussion has been modified accordingly.
Minor:
Line 101-103: How many data sets were excluded?
As reported in the Results section, placental volume has been assessed in 65 high-quality out of 98 datasets (measurement success rate 66%).
- 135: the correct biological term is sex rather than gender. Please change troughout.
The text has been modified as suggested.
- L. 205f: leave out the word "significant" and include p-value.
The text has been modified as suggested.
- 280ff: A sentence of more than 7 lines may be a bit too long.
The text has been modified as suggested.
Reviewer 2 Report
Nutrients-816822 Associations between first trimester maternal nutritional score, early markers of placental function and pregnancy outcome
This certainly is an interesting manuscript describing a pattern of food intake with early biomarkers of pregnancy and pregnancy outcomes. The work in general has been performed well with some minor issues identified below.
Major comments
- Very well written and logical introduction.
- Methods: is the first trimester combined screening test for aneuploidies offered to all pregnant women in Italy or is there a risk stratification?
- Methods: some of the exclusion criteria such as IUFD and congenital anomalies could only be applied after the outcome of pregnancy is known—have the authors investigated if there were differences in adherence to the Mediterranean diet between women who did or did not have IUFD or congenital anomalies?
- Methods, data collection: for how many participants were the data on birth outcomes supplied by the mother rather than from the charts?
- Methods, placental volume determination: were these measurements adjusted for gestational age? Similarly, were the biochemical values adjusted for gestational age? This could significantly affect the results because of the rapid growth in the placenta between weeks 10 to 14.
- Methods, nutritional questionnaire: the component fruit and vegetables, yes has a frequency of 40%. Do I interpret this correctly as only 40% of women had intake of >=5 serves/day or should I interpret this as only 40% of women ate fruits and vegetables (which is unlikely). It may be good to clarify this.
- Methods, statistics: why was LGA not included as pregnancy outcomes that were associated with first trimester nutritional score? Were personal or family history for preterm birth, hypertensive disorders and SGA adjusted for in the analyses?
- Results, table 2: how many of the women were underweight e.g. BMI<18.5? This could affect the expected rates of SGA. Similarly, how many women were overweight or obese?
- Results, table 2: was the GWG measured as weight gain from prepregnancy weight to weight at the end of the first trimester? How was prepregnancy weight established? Can GWG over the full range of gestation also be included?
- Results, table 2: PAPP-A are often presented as MoM (multiple of median), can this be added to the table?
- Results, nutritional score: was the nutritional score related to BMI at all?
- Results, table 3: 203 days gestation is only 29 weeks gestation: How many babies were born preterm and can you specify how many were born <32+0 weeks; <36+6 weeks?
- Results, table 3: can you specify which hypertensive disorders were present and if these were related to the preterm deliveries at all. Because it is possible that the preterm birth was due to severe preeclampsia which is different from a spontaneous preterm birth or a preterm birth due to FGR or fetal distress.
- Results, table 3: it is obvious from table 3 that GDM was one of the pregnancy outcomes that was investigated. Why is this not highlighted earlier in the manuscript or in the abstract? And why is it not included in the association study or are the results of that not mentioned?
- Discussion, negative association between placental volume and adherence to the Mediterranean diet: this is surprising. The authors suggest that a smaller placental volume but a high CRL/placental volume ratio indicates that the placenta is more efficient. If that is the case, there should be a relationship between placental volume and birth weight: was this relationship evaluated. If not, it should be done just to provide additional support for the hypothesis of the authors. Given that there is no adjustment for gestational age, I am wondering if GA would be an alternative explanation for the negative association reported. The period between 11 and 14 weeks gestation also comprises the period where there is remodelling of the spiral arteries which may affect vascular resistance in the placenta affecting UtA doppler indices.
Author Response
Reviewer 2
Comments and Suggestions for Authors
Nutrients-816822 Associations between first trimester maternal nutritional score, early markers of placental function and pregnancy outcome
This certainly is an interesting manuscript describing a pattern of food intake with early biomarkers of pregnancy and pregnancy outcomes. The work in general has been performed well with some minor issues identified below.
Major comments
Very well written and logical introduction.
We appreciate the reviewer's comment.
Methods: is the first trimester combined screening test for aneuploidies offered to all pregnant women in Italy or is there a risk stratification?
The first trimester combined screening is offered to all pregnancies, independently on a priori risk. This important information has been added to the M&M section.
Methods: some of the exclusion criteria such as IUFD and congenital anomalies could only be applied after the outcome of pregnancy is known—have the authors investigated if there were differences in adherence to the Mediterranean diet between women who did or did not have IUFD or congenital anomalies?
We thank the reviewer for this comment: actually, no IUFD have been detected in the study population, whereas three congenital anomalies have been excluded for in utero diagnosis, further confirmed after birth (n=1 clubfoot, n=1 hydronephrosis, n=1 interventricular septal defect). Despite the very small sample does not allow to speculate on this association, no differences were detected in nutritional score distribution between pregnancies excluded for congenital anomalies and the included study population. This information has been added to the Results section.
Methods, data collection: for how many participants were the data on birth outcomes supplied by the mother rather than from the charts?
Birth outcomes were collected through phone interview in 25 out of 94 patients, 27 patients sent the hospital discharge letter by email, and for the remaining 42 patients birth data were directly collected from hospital registries.
Methods, placental volume determination: were these measurements adjusted for gestational age? Similarly, were the biochemical values adjusted for gestational age? This could significantly affect the results because of the rapid growth in the placenta between weeks 10 to 14.
We completely agree with the reviewer and both crude and fully adjusted models include adjustment for gestational age (as specified in the Statistical analysis section and table 4).
Methods, nutritional questionnaire: the component fruit and vegetables, yes has a frequency of 40%. Do I interpret this correctly as only 40% of women had intake of >=5 serves/day or should I interpret this as only 40% of women ate fruits and vegetables (which is unlikely). It may be good to clarify this.
We apologize for this unclear information: the answer 'yes' means that the woman eats at least 5 portions of fruit/vegetables per day, as required by the national nutritional guideline for a healthy pregnancy. This was specified in table 2 footnotes and it is now better described in the M&M section.
Methods, statistics: why was LGA not included as pregnancy outcomes that were associated with first trimester nutritional score? Were personal or family history for preterm birth, hypertensive disorders and SGA adjusted for in the analyses?
We thank the reviewer for this very relevant comment. The study population includes 9 babies classified as LGA (birth weight >90th percentile for gestational age) and the association with maternal nutritional score has been evaluated, with no significant results. We have modified the M&M and Results section according to this suggestion.
With regard to the second point raised by the reviewer, unfortunately personal and family history of obstetric complications was not collected, so that we have now added this as a limitation of the study.
Results, table 2: how many of the women were underweight e.g. BMI<18.5? This could affect the expected rates of SGA. Similarly, how many women were overweight or obese?
In line with the reviewer 1' suggestion, this important information has been added to the Results section. The study population included 11 patients being underweight and seven obese, in line with national prevalence of abnormal BMI among fertile women. Pregestational BMI was not correlated to maternal nutritional score, probably as an effect of an early nutritional counselling addressed to all women with abnormal BMI, as required by local obstetric care. In a multi-adjusted model, no associations were detected between pregestational BMI and birth outcomes, including birthweight and gestational age at birth.
Results, table 2: was the GWG measured as weight gain from prepregnancy weight to weight at the end of the first trimester? How was prepregnancy weight established? Can GWG over the full range of gestation also be included?
GWG was recorded as weight gain between pregestational weight as reported by the woman and maternal weight at enrollment. GWG at term was not available and this point was discussed as a limitation of the study.
Results, table 2: PAPP-A are often presented as MoM (multiple of median), can this be added to the table?
Table 2 has been modified as required.
Results, nutritional score: was the nutritional score related to BMI at all?
We thank the reviewer for this question and -also in response to reviewer 1- this analysis has been added to the Results section.
Results, table 3: 203 days gestation is only 29 weeks gestation: How many babies were born preterm and can you specify how many were born <32+0 weeks; <36+6 weeks?
One baby was delivered before 32 weeks of gestation (29 weeks), and 5 babies were spontaneously delivered between 32 and 37 weeks. This information has been added to the Results section.
Results, table 3: can you specify which hypertensive disorders were present and if these were related to the preterm deliveries at all. Because it is possible that the preterm birth was due to severe preeclampsia which is different from a spontaneous preterm birth or a preterm birth due to FGR or fetal distress.
We completely understand the reviewer's comment. Gestational hypertension was diagnosed among three included women and underwent vaginal delivery at term. Six spontaneous preterm delivery were recorded and classified as previously specified. Table 3 has been modified by adding "spontaneous" and hypertensive disorders further specified as required.
Results, table 3: it is obvious from table 3 that GDM was one of the pregnancy outcomes that was investigated. Why is this not highlighted earlier in the manuscript or in the abstract? And why is it not included in the association study or are the results of that not mentioned?
We agree with the reviewer and further included this analysis. Again, no associations were detected between first trimester nutritional score and GDM.
Discussion, negative association between placental volume and adherence to the Mediterranean diet: this is surprising. The authors suggest that a smaller placental volume but a high CRL/placental volume ratio indicates that the placenta is more efficient. If that is the case, there should be a relationship between placental volume and birth weight: was this relationship evaluated. If not, it should be done just to provide additional support for the hypothesis of the authors. Given that there is no adjustment for gestational age, I am wondering if GA would be an alternative explanation for the negative association reported. The period between 11 and 14 weeks gestation also comprises the period where there is remodelling of the spiral arteries which may affect vascular resistance in the placenta affecting UtA doppler indices.
We very much appreciate and understand this comment, as this result was surprising to us as well. Actually, the adjustment for GA at the time of placental volume measurement was performed. We additionally performed the multivariate analysis as suggested by the reviewer and found the positive association between placental volume and birth weight, which corroborates our hypothesis.
Round 2
Reviewer 2 Report
Thanks to the authors for their thorough revision of the manuscript. I have no further comments.